# The genomic footprint of social stratification in admixing American populations

Alex Mas-Sandoval[1,2]*, Sara Mathieson[3], Matteo Fumagalli[1,4]*

[1]Department of Life Sciences, Silwood Park Campus, Imperial College London, London, United Kingdom; [2]Department of Statistical Sciences, University of Bologna, Bologna, Italy; [3]Department of Computer Science, Haverford College, Haverford, United States; [4]School of Biological and Behavioural Sciences, Queen Mary University of London, London, United Kingdom

**Abstract** Cultural and socioeconomic differences stratify human societies and shape their genetic structure beyond the sole effect of geography. Despite mating being limited by sociocultural stratification, most demographic models in population genetics often assume random mating. Taking advantage of the correlation between sociocultural stratification and the proportion of genetic ancestry in admixed populations, we sought to infer the former process in the Americas. To this aim, we define a mating model where the individual proportions of the genome inherited from Native American, European, and sub-Saharan African ancestral populations constrain the mating probabilities through ancestry-related assortative mating and sex bias parameters. We simulate a wide range of admixture scenarios under this model. Then, we train a deep neural network and retrieve good performance in predicting mating parameters from genomic data. Our results show how population stratification, shaped by socially constructed racial and gender hierarchies, has constrained the admixture processes in the Americas since the European colonization and the subsequent Atlantic slave trade.

**\*For correspondence:**
alex.massandoval@unibo.it (AM-S);
m.fumagalli@qmul.ac.uk (MF)

**Competing interest:** The authors declare that no competing interests exist.

## Editor's evaluation

In this important study, the authors develop a neural network to investigate assortative mating and sex-bias in admixed populations from the Americas. Applying their method to modern-day human genomes, they estimate sex-biased admixture and ancestry-based assortative mating. The evidence supporting their claims is solid, and their results will be of interest to population geneticists, anthropologists, and those interested in the history of the Americas.

## Introduction

### Ancestry-related assortative mating and sex bias in admixture models

Assortative mating is the phenomenon whereby mates resemble each other more than would occur under random mating. Through the lens of population genetics, assortative mating entails the correlation of genetic variants between mates (*Versluys et al., 2021*). Assortative mating is often a consequence of a subdivided population structure, observed in most human populations (*Cavalli-Sforza and Feldman, 1981*; *Nagoshi et al., 1990*; *Sebro and Risch, 2012*; *Sebro et al., 2017*). In many species, assortative mating is also the product of active mate choice, an adaptive behavior by which individuals choose a genetically similar mate by their phenotype (*Merrill et al., 2019*; *Versluys et al., 2021*). Whether or not active mate choice significantly takes place in human populations is unclear,

as it is challenging to discern the effect it might have beyond that of population structure (*Eshel and Cavalli-Sforza, 1982*; *Xie et al., 2015*; *Sebro and Risch, 2012*; *Abdellaoui et al., 2014*; *Sebro et al., 2017*).

Geography structures populations of most species, including humans. Individuals separated by shorter distances interact more than individuals separated by longer distances or geographical features like watercourses or mountain ranges, which imply limited mating as the geographical barriers increase (*Wright, 1943*; *Malecot, 1948*; *Kimura and Weiss, 1964*; *Cavalli-Sforza et al., 1996*; *Novembre et al., 2008*). The reduced gene-flow between geographical groups leads to differentiated processes of genetic drift and results in distinguishable genetic profiles known as genetic ancestries (*Mathieson and Scally, 2020*; *Lewis et al., 2022*; *Coop, 2022*).

In human populations, culture also shapes the population structure. Socioeconomic and cultural barriers, which might have a certain permeability, limit the interaction between human groups. These social groups with a common language, religion, socioeconomic status, etc. define overlapping subpopulations where mating takes place mostly within them (*Manni, 2010*; *Campbell, 2015*; *Matsumae et al., 2021*).

In the case of migration, two or more populations cohabit in the same location and eventually admix to become subpopulations of a newly admixed population. The admixture process does not take place randomly but it is constrained by barriers set by socioeconomic and cultural differences between subpopulations with their own distinguishable profile of genetic ancestries (*Risch et al., 2009*; *Nagoshi et al., 1990*; *Sebro et al., 2017*). These barriers are socially constructed and, particularly in colonial contexts, their permeability is often politically restricted (*McLean, 2021*).

In admixed individuals, the genetic ancestry related to each source population can be tracked along the genome and expressed in individual-based proportions.

Therefore, in recently admixed populations, the population structure driven by culture and socioeconomic differences is associated with differences in the proportions of genetic ancestry. As a consequence, the proportion of genetic ancestry between mates correlate. This phenomenon is defined as ancestry-related assortative mating (*Burrell and Disotell, 2009*; *Bryc et al., 2010*; *Norris et al., 2019*). In addition, individuals might show a preference towards mating partners of the opposite sex with lower or higher ancestry proportions, which is defined as ancestry-related sex bias (*Goldberg and Rosenberg, 2015*).

In admixed populations, the length of the continuous ancestry tracts is widely used to infer the time since admixture under the assumption of random mating. During gametogenesis in admixed individuals, recombination breaks down continuous ancestry tracts inherited from each of the source populations of the admixture event into smaller alternate fragments at each generation. Consequently, the length of the continuous ancestry tracts reflects how many generations ago the source populations migrated across the geographical barriers that prevented them to mate (*Gravel, 2012*; *Hellenthal et al., 2014*; *Chintalapati et al., 2022*). Herein, we argue that the tract length information can measure the non-randomness of mating associated with genetic ancestry and, therefore, it can also monitor the permeability of socioeconomic and cultural barriers between subpopulations with different genetic ancestries (*Zaitlen et al., 2017*).

Some of the methods to date admixture can discern multiple pulses of migration. Only a few of them have addressed complex admixture histories such as the fluctuation of unbalanced migrations of males and females from two source populations (*Laurent et al., 2022*). However, almost all these approaches assume random mating in the admixed population, overlooking the effect of population stratification in the population structure. To our knowledge, few studies have modeled ancestry-related assortative mating during admixture, although limited to two source populations (*Goldberg et al., 2020*; *Kim et al., 2021*).

Beyond analytical modeling, population genetics studies have also measured ancestry-related assortative mating through the correlation of genetic ancestry proportions between mates (*Bryc et al., 2010*; *Korunes et al., 2022*; *Arauna et al., 2022*). Non-random mating can also be monitored through deviations of the observed heterozygosity from Hardy-Weinberg equilibrium expected values (*Crow and Felsenstein, 1968*). Thus, when information on genetic ancestry of mating couples is not available, it is still possible to infer ancestry-related assortative mating through the comparison of the genetic ancestry of the two homologous chromosomes of the individuals (*Norris et al., 2019*). However, these approaches can only infer the mating patterns from the last generations.

Despite these efforts, we still lack a rigorous and robust method to shed light onto the patterns of ancestry-related non-random mating across all generations in which the admixture process extends. More specifically, we are in need of a comprehensive model of ancestry-related assortative mating and sex bias, two parameters which have been rarely jointly modeled in population genetics.

## Social stratification and population structure in the Americas

Among human populations, the admixing populations from the Americas are of special interest in admixture studies. We consider them as *admixing* populations, because their genetics is shaped by an ongoing admixture process of three differentiated continental ancestries that started five centuries ago, constrained by a strong social structure.

At the end of the 15[th] century, European powers initiated the colonization process in the lands inhabited by Native Americans. In this frame, European colonizers enslaved more than 10 million people brought from sub-Saharan Africa (*Eltis, 2018*). As a result of this historical event, the populations of the Americas are the outcome of the admixture of Native American, European, and sub-Saharan genetic ancestries (*Salzano and Bortolini, 2005*; *Bedoya et al., 2006*; *Wang et al., 2008*; *Moreno-Estrada et al., 2013*; *Gravel et al., 2013*; *Ruiz-Linares et al., 2014*; *Adhikari et al., 2017*; *Adhikari et al., 2016*; *Ongaro et al., 2019*).

After the abolition of slavery, most of these populations remained stratified based on the socioeconomic status and according to hierarchical notions of racial difference. Some of them have even experienced explicit segregation policies long after the abolition that aimed to prevent mating between subpopulations from different origins and maintain socioeconomic stratification (*Douglass, 1882*; *Du Bois, 1935*; *Davis, 1981*).

In Latin America, in addition to segregation, European colonial powers and creole elites implemented eugenicist policies under the frame of *mestizaje/mestiçagem. Mestizaje/mestiçagem* refers to the process of admixture of Native American, European, and sub-Saharan ancestries in the context of the European colonization. It is, therefore, associated to the mixture across hierarchical differences understood as 'racial,' differences of class, and differences of gender. Since mid-nineteenth century, Latin American nation-building elites have aimed to associate *mestizaje/mestiçagem* to an equalizing process, by claiming that it overcomes and blurs the socioeconomic differences related to 'race.' However, critics have argued that the *mestizaje/mestiçagem*'s notion of hybridity inherently entails the idea of its constitutive origins and the hierarchies that order those origins. In this sense, *mestizaje/mestiçagem* attaches greater value to the interactions that move towards whiteness and masculinity and lower value to those that move towards blackness or indigeneity, and femininity (*Wade, 2017*; *Wade et al., 2020*; *Abel, 2022*).

By analyzing the impact of the European colonization in the population structure through mating, we aim to evaluate the stratification related to the genetic ancestry not only quantifying the population subdivision but also measuring the genetic ancestry asymmetry between males and females in mating. Following this approach, we conceptualize a novel mechanistic mating model that explicitly integrates ancestry-related assortative mating and sex bias jointly, through an intersectional approach derived from the interrelated hierarchies observed in the admixture process (*Crenshaw, 1989*; *Crenshaw, 1991*; *Wade, 2017*). We consider a three-way admixture scenario mirroring the demography of the admixing American populations. We build and train a deep neural network to infer non-random mating parameters using extensive synthetic data. We deploy this network to genomic data from admixing American populations sequenced as part of the 1000 Genomes Project 1000 (*Auton et al., 2015*) and quantify the extent of ancestry-related assortative mating and sex bias. Finally, we discuss racial and gender hierarchies as inferred from their footprint on genetic structure.

## Results and discussion

We report our results in three sections: (i) the novel mating model and framework for simulations, (ii) the performance of the neural network, and (iii) the inference of the ancestry-related mating probabilities for admixing American populations.

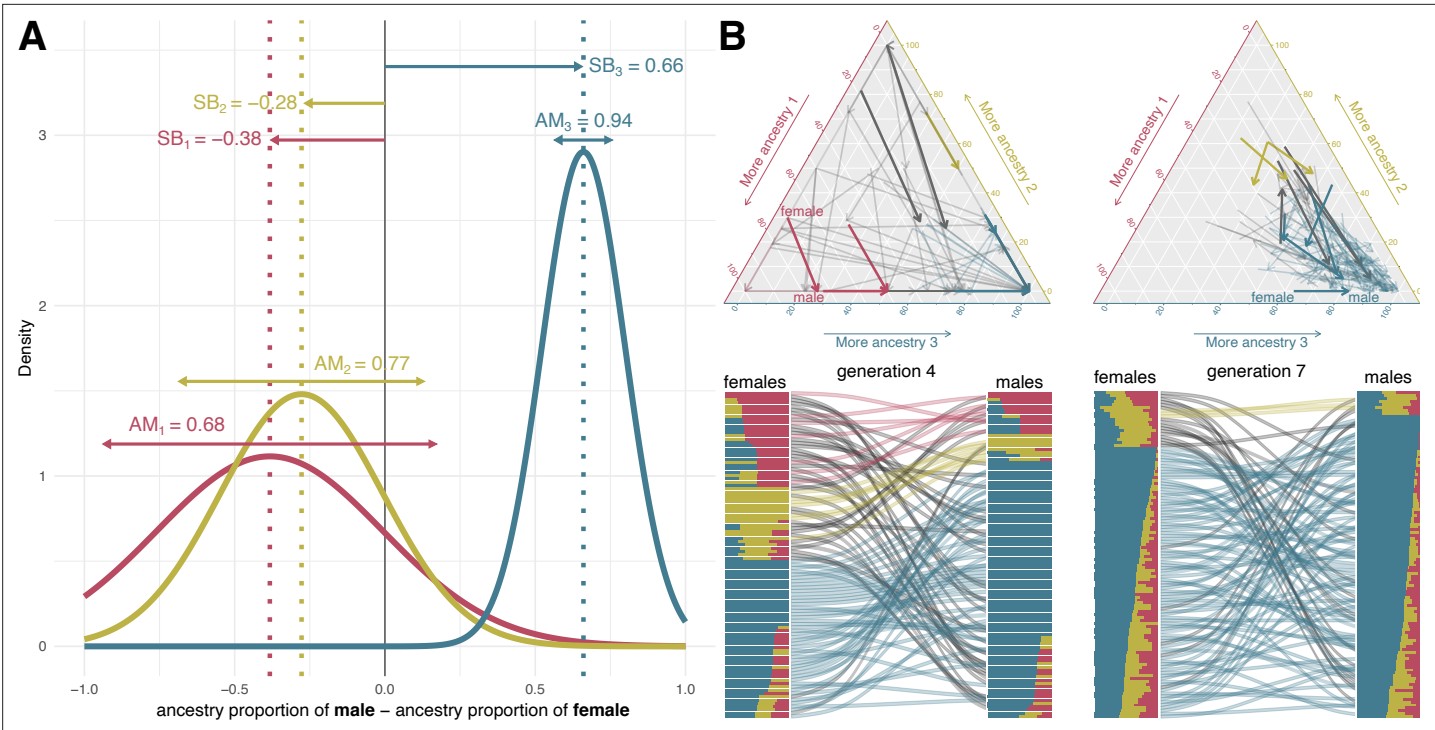

**Figure 1.** Mating model. (**A**) Assortative mating (AM) and sex bias (SB) values that modulate the mating probabilities in a simulation example of 19 generations from the colonization of America to nowadays. The mating probability for a given couple is set as a function of the differences in the genetic ancestry proportions for each ancestry. We assume the mating probability follow a three-dimensional normal distribution. In this normal distribution, SB sets the expected value and AM is inversely proportional to its variance. (**B**) Ancestry proportions of mating couples at generations 4 and 7 in ternary plots (top) and barplots (bottom) based on the mating probabilities defined in A. In the top plots, each arrow represents a couple. The arrow tail and head coordinates in the ternary plots show the ancestry proportions of the female and the male, respectively. In the bottom, the barplots represent male and female ancestry proportions, linked by curved lines reflecting mating. Red, yellow, and blue correspond to ancestries 1, 2, and 3. The arrows in the ternary plot and the lines between barplots representing a mating couple are colored with the color corresponding to the predominant ancestry in both male and female, and they are depicted in black if it differs between them.

The online version of this article includes the following figure supplement(s) for figure 1:

**Figure supplement 1.** Admixture Model for One Pulse and Two Pulses.

## An ancestry-related mating model

We present an admixture model defined by the mating probabilities of all possible male and female couples, set by their ancestry proportion difference. For each ancestry, the ancestry-related sex bias (SB) and the ancestry-related assortative mating (AM) parameters determine the mating probability of each couple as a function of the difference in the ancestry proportion between male and female. We assume that the differences in the ancestry proportions within the mating couples follow a Normal distribution that translates into the mating probabilities (*Figure 1*).

The expected value of this Normal distribution defines SB, while AM is modeled as being inversely proportional to the variance (see Methods and materials for mathematical details). For a given ancestry, positive SB values indicate that couples where males have a higher proportion than females of this ancestry have more chances of mating, with the opposite pattern for negative SB values. Conversely, AM modulates the decay of mating probability when the difference of ancestry proportion within the couple moves away from the expected value set by the SB parameter. Therefore, when SB is close to zero and AM is high, a couple with similar ancestry proportions have much higher probability of mating than a couple with substantial differences in the ancestry proportions. If AM is low, a couple with similar ancestry proportions only have a slightly higher probability to mate than a couple with substantial differences in the ancestry proportions. The AM parameter measures the non-randomness of mating associated to a genetic ancestry. This includes both *positive assortative mating* -genetic similarity between mates- (when SB is zero)

and *negative assortative mating* -genetic dissimilarity between mates- (when *SB* is not zero). This approach allows accounting for the male-female way of negative assortative mating through SB parameter. *Figure 1B* shows how a sample of individuals in generations 4 and 7 mate based on the mating probabilities set by the example values of AM and SB defined in *Figure 1A* and *Figure 1— figure supplement 1*.

We focus on the case of three-way ancestry, a model that describes the admixture of the populations of the Americas and their triple genetic ancestry: Native American, European, and sub-Saharan African. We define two alternative models, referred to as the One Pulse model and the Two Pulses model. The simpler One Pulse model assumes one migration event occurring 19 generations ago and includes five independent parameters: $AM_1$, $AM_2$, $AM_3$, $SB_1$, and $SB_2$ for sub-Saharan African (1), Native American (2), and European (3) ancestries (*Figure 1A*). In the Two Pulses model, an additional parameter (the Gene Flow Rate 9 generations ago -GFR-) determines the fraction of the gene pool arriving in a second migration from each source population 9 generations ago (at generation 10) (*Figure 1— figure supplement 1*). In both models we assume a continuous admixture process that starts 19 generations ago, knowing that the populations analysed trace the first contact of Native American and European populations in the first half of 16th century (*Sánchez-Albornoz, 1977*; *Thornton, 1987*) and assuming a generation time of 26.9 years (*Wang et al., 2023*). In contrast with the approaches that aim to find an admixture date assuming random mating, we assume that the admixture process starts with the contact, and it is continuous and modulated with the mating parameters that we aim to infer.

Our goal is to predict AM and SB (and GFR for the Two Pulses model) for the admixing American populations sampled in the 1000 genomes project (African Caribbeans in Barbados, ACB; African Ancestry in South-West USA, ASW; Colombians in Medellín, CLM; Mexicans in Los Angeles, MXL; Peruvians in Lima, PEL; and Puerto Ricans in Puerto Rico, PUR). To do so, we aim to compare the continuous ancestry tract lengths profile obtained from a local ancestry analysis performed on this data to the tract lengths profile issued from simulated data for each population with known combinations of the mating parameters.

For 10,000 random combinations of AM and SB parameters (and GFR for the Two Pulses model) for each population we simulate, forward-in-time, a range of admixture scenarios. The contribution of each genetic ancestry to the gene pool of the simulated admixed population is equivalent to the observed ancestry proportions after the local ancestry analysis on the real data. We simulate 22 autosomal chromosomes and the X chromosome for each individual at each generation, keeping track of the local genetic ancestry at each chromosomal region (*Figure 1*, *Figure 2A*). This approach serves a dual purpose: (i) to simulate the mating as a function of the genome-wide ancestry proportions of all males and females, based on the mating probabilities set by the AM and SB parameters of the mating model (*Figure 1*); (ii) to generate the continuous ancestry tract lengths profile as an output after the last simulated generation, which counts the number of fragments within each of the 22 windows defined by a length interval in cM, in a logarithmic scale (*Figure 2B*; see Materials and methods).

## A deep neural network to estimate mating parameters efficiently

To infer all parameters in our model, we train a deep neural network for each population. By exploring the entire parameter space of AM and SB parameters (and for the Two Pulses model), we feed simulated continuous ancestry tract length profiles to a deep neural network consisting of fully-connected layers (*Figure 2C*).

The network sufficiently learn the weights for all parameters without overfitting over 40 epochs, as shown by the decay of the loss function (mean squared error) (*Figure 3—figure supplement 1*). We observe low mean squared error on the testing set for all parameters (*Figure 3*). Similarly, we appreciate a high correlation between true and predicted values, as shown by $R^2$ values and the confusion matrices, at testing (*Figure 3—figure supplement 2*).

The trained network exhibits better predictions for AM parameters than for an parameters across all ancestries, populations, and migration models. Interestingly, the higher complexity of the Two Pulses migration model does not produce a higher mean squared error or lower $R^2$ values for any of the tested parameters. In fact, the mean of the mean squared error for the Two Pulses model is only marginally higher than the mean of the mean squared error for the simpler One Pulse model (*Figure 3*, *Figure 3—figure supplement 1*, *Figure 3—figure supplement 2*).

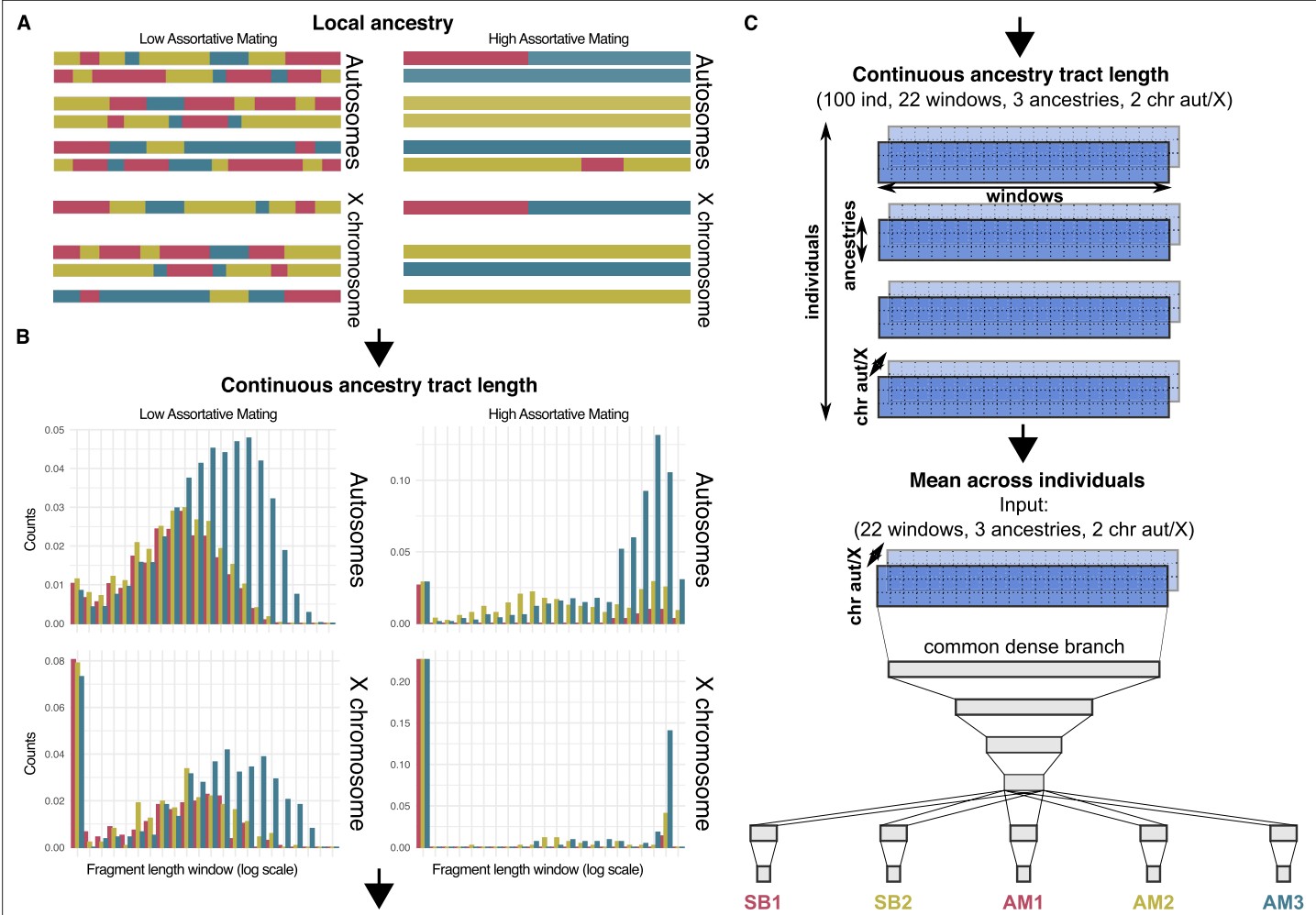

**Figure 2.** Local ancestry, Contiuous ancestry tract length and Neural network. (**A**) Schematic view of the autosomal and sex chromosomes split into the continuous ancestry tracts inherited from each of the three ancestries after a local ancestry analysis with RFMix. (**B**) Continuous ancestry tract length profile displaying the number of tracts for each ancestry in each tract length bin. The break points that define the bin widths are set in a logarithmic scale. (**C**) Matrix representing the continuous ancestry tract length profile accounting for the amount of tracts in each length bin, for each ancestry in either autosomal or sex chromosome for each individual. The mean across individuals summarises the four-dimensional matrix in a population three-dimensional matrix, which is used as the input of the neural network. The neural network has four fully connected layers that split into a branch for each parameter, each one made of a last hidden layer connected to the output layer.

The online version of this article includes the following figure supplement(s) for figure 2:

**Figure supplement 1.** Individual proportions of sub-Saharan (red), Native American (green), and European (blue) ancestry were inferred after a Local Ancestry analysis with RFMix for autosomes (Aut), on the left, and X chromosome (X), on the right, for each population.

**Figure supplement 2.** Individual proportions of sub-Saharan (red), Native American (green), and European (blue) ancestry inferred after a Local Ancestry analysis with Gnomix for autosomes (Aut), on the left, and X chromosome (X), on the right, for each population.

**Figure supplement 3.** Distribution of individual proportions of sub-Saharan (red), Native American (green), and European (blue) ancestry inferred after a Local Ancestry analysis with RFMix for autosomes (Aut), on the left, and X chromosome (X), on the right, for each population. The box limits are the 25th and 75th percentiles and the points show the outliers 1.5 times the interquartile range above the 75th percentile and below the 25th percentile.

**Figure supplement 4.** Distribution of individual proportions of sub-Saharan (red), Native American (green), and European (blue) ancestry inferred after a Local Ancestry analysis with Gnomix for autosomes (Aut), on the left, and X chromosome (X), on the right, for each population. The box limits are the 25th and 75th percentiles and the points show the outliers 1.5 times the interquartile range above the 75th percentile and below the 25th percentile.

**Figure supplement 5.** Continuous ancestry tract lengths profile showing the amount of fragments in each length window for sub-Saharan (red), Native American (green), and European (blue) ancestry for each population, after a local ancestry analysis with RFMix.

**Figure supplement 6.** Continuous ancestry tract lengths profile showing the amount of fragments in each length window for sub-Saharan (red), Native American (green), and European (blue) ancestry for each population, after a local ancestry analysis with Gnomix.

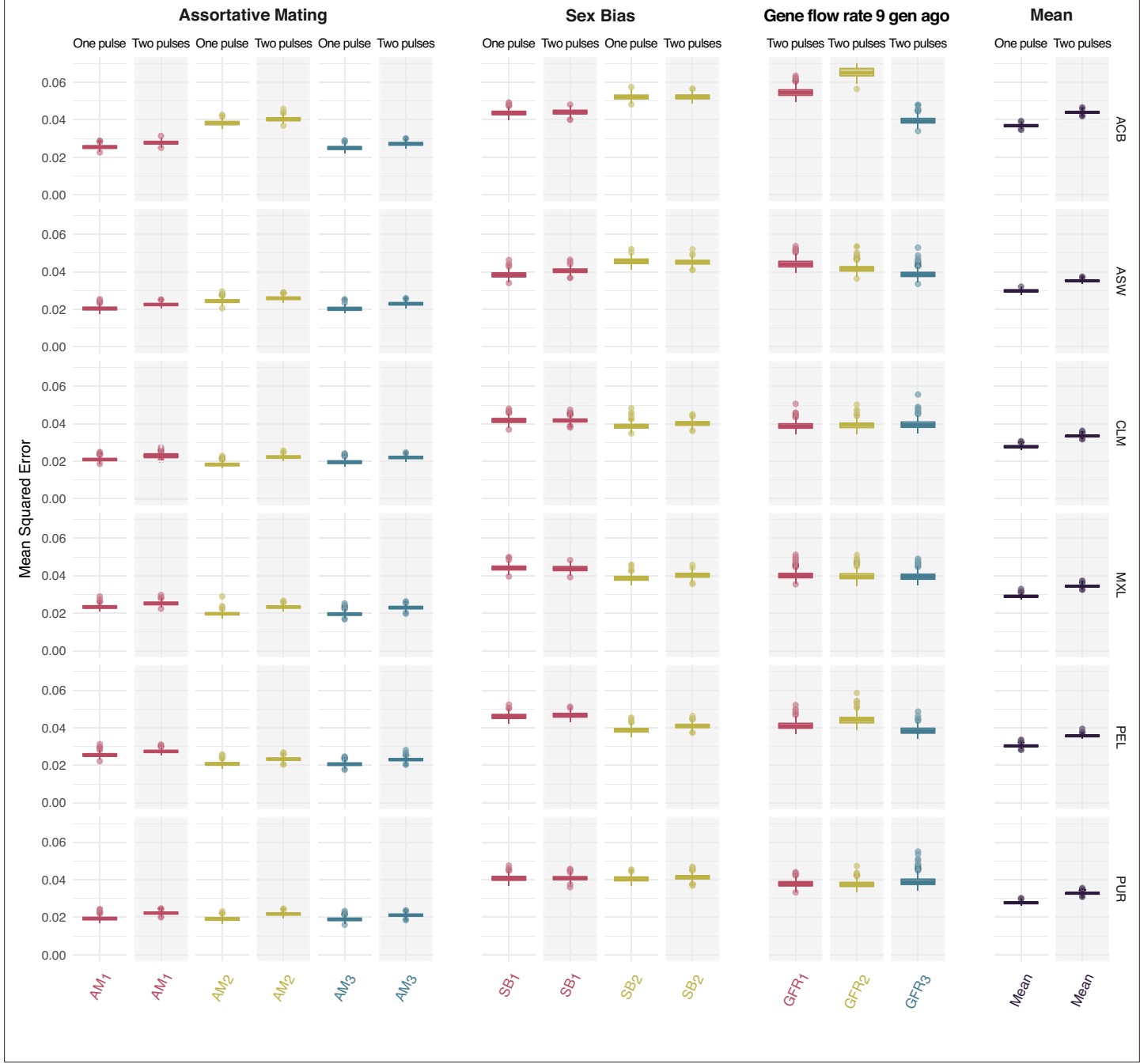

**Figure 3.** Mean squared error comparing true and predicted values at testing for the *assortative mating*, *sex bias*, *gene flow rate 9 generations ago* parameters for each ancestry for both One Pulse and Two Pulses models and mean values for each model. Each color represents the values for a different ancestry (red, yellow, and blue for ancestries 1, 2, and 3 respectively, which correspond to sub-Saharan, Native American and European ancestries). The boxplot shows the distributions of values across the 1000 trained neural networks. The box limits are the 25th and 75th percentiles and the points show the outliers 1.5 times the interquartile range above the 75th percentile and below the 25th percentile.

The online version of this article includes the following figure supplement(s) for figure 3:

**Figure supplement 1.** Loss function at training for the *assortative mating*, *sex bias*, *gene flow rate 9 generations ago* parameters for each ancestry for both One Pulse and Two Pulses models and mean values for each model.

**Figure supplement 2.** Confusion matrices and $R^2$ comparing true (x-axis) and predicted (y-axis) values at testing for *assortative mating*, *sex bias*, *gene flow Rate 9 generations ago* parameters for each ancestry for both One Pulse and Two Pulses models.

The performance of the Local Ancestry analysis might constrain the shape of the continuous ancestry tract length profile and the accuracy of the predictions of the mating parameters. RFMix has been the state-of-the-art method for Local Ancestry method during the last decade. Recently, Gnomix gained popularity after claiming much lower computational requirements and similar or even better accuracy (*Hilmarsson et al., 2021*). We performed a Local Ancestry analysis with Gnomix and we generated the continuous ancestry tract lengths profile after it. The Gnomix tract length profile showed higher values in the shortest tracts window than the RFMix one (*Figure 2—figure supplements 1–6*). *Gravel, 2012* alerted that short ancestry tracts are likely to have reduced accuracy rates in Local Ancestry inference, whereas longer tracts can be detected with increased confidence.

To evaluate possible biases introduced by the wrong assignation of isolated windows in the Local Ancestry inference, we tested the performance and the estimations of a Neural Network using a modified tract lengths profile for both RFMix and Gnomix. We modified the tract lengths profile by removing, or not, the window corresponding to the shortest tracts, or by dividing, or not, each value of the histogram by the total number of tracts in the Autosomes and in the X Chromosome. The tract lengths profile with *All windows* and *Divided by total sum of tracts* presents a low value of generalized variance (GV) between RFMix and Gnomix estimations and a low MSE value in the testing of the Neural Network (Tables 7 and 8, *Figure 4—figure supplements 3–7*). This shows that a matrix normalization of the tract length profile is enough to reduce the effects of the eventual biases introduced by the Local Ancestry method and points that the Neural Network is able to capture relevant information from the profile shape.

## The Native American and sub-Saharan genetic ancestries respectively shape the mating probabilities in Latin American and African American populations

We sought to test the occurrence and extent of assortative mating and sex bias in the admixing American populations from 1000 genomes. To predict AM and SB parameters (and GFR for the Two Pulses model) we deployed the trained neural network on the continuous ancestry tract length profiles of these populations were obtained after a local ancestry analysis.

In the One Pulse model, the Latin American populations (CLM, MXL, PEL, PUR) present a consistent pattern where the Native American ancestry shapes the mating probabilities, as the AM parameter associated to this ancestry is the highest in all populations. Thus, the differences in the Native American ancestry between males and females modulate the mating in Latin American populations, although the Native American ancestry is not the one observed in highest proportion in all of them. PEL, CLM, and MXL populations show stronger AM values than PUR. The high AM values are coupled with negative SB for CLM and PUR populations, indicating that females of high Native American ancestry are more likely to mate with males of lower Native American ancestry. CLM and PUR exhibit significant negative sex-biased admixture while MXL and PEL do not. Conversely, ASW population presents the highest AM in the sub-Saharan African ancestry. Paired with a positive SB value, these estimates indicate that males of high sub-Saharan African ancestry are more likely to mate with females of lower sub-Saharan African ancestry. Finally, ACB populations show similar AM values for the three ancestries, with no specific ancestry modulating the mating probability (Table 3, *Figure 4A*).

In the Two Pulses model, we allow for an additional migration pulse 9 generations ago (at generation 10) and we let the neural network predict the gene flow rate through the GFR parameter. Under this new scenario, as expected, AM values are much lower than their corresponding values under a One Pulse model. In fact, part of the population structure that is modeled as social stratification in the One Pulse model is now modeled by gene flow from an additional migration event. Both models reflect similar admixture dynamics, where Native American and sub-Saharan African genetic components take longer to homogenize across the individuals of Latin American and African American populations, respectively (*Figure 4B*).

Under both models of migration, the effect that SB has on the mating probabilities depends on the AM values, as lower AM values imply a lower effect of SB on the mating. In case of low AM, individuals are less constrained in their mating by their ancestry and, therefore, the effect of SB is less prominent.

To evaluate the similarity of the footprints left by either assortative mating or gene flow due to migration, we tested how a neural network trained to predict GFR could predict GFR from data with no gene-flow due to migration but only assortative mating. And, in parallel, we tested how a neural

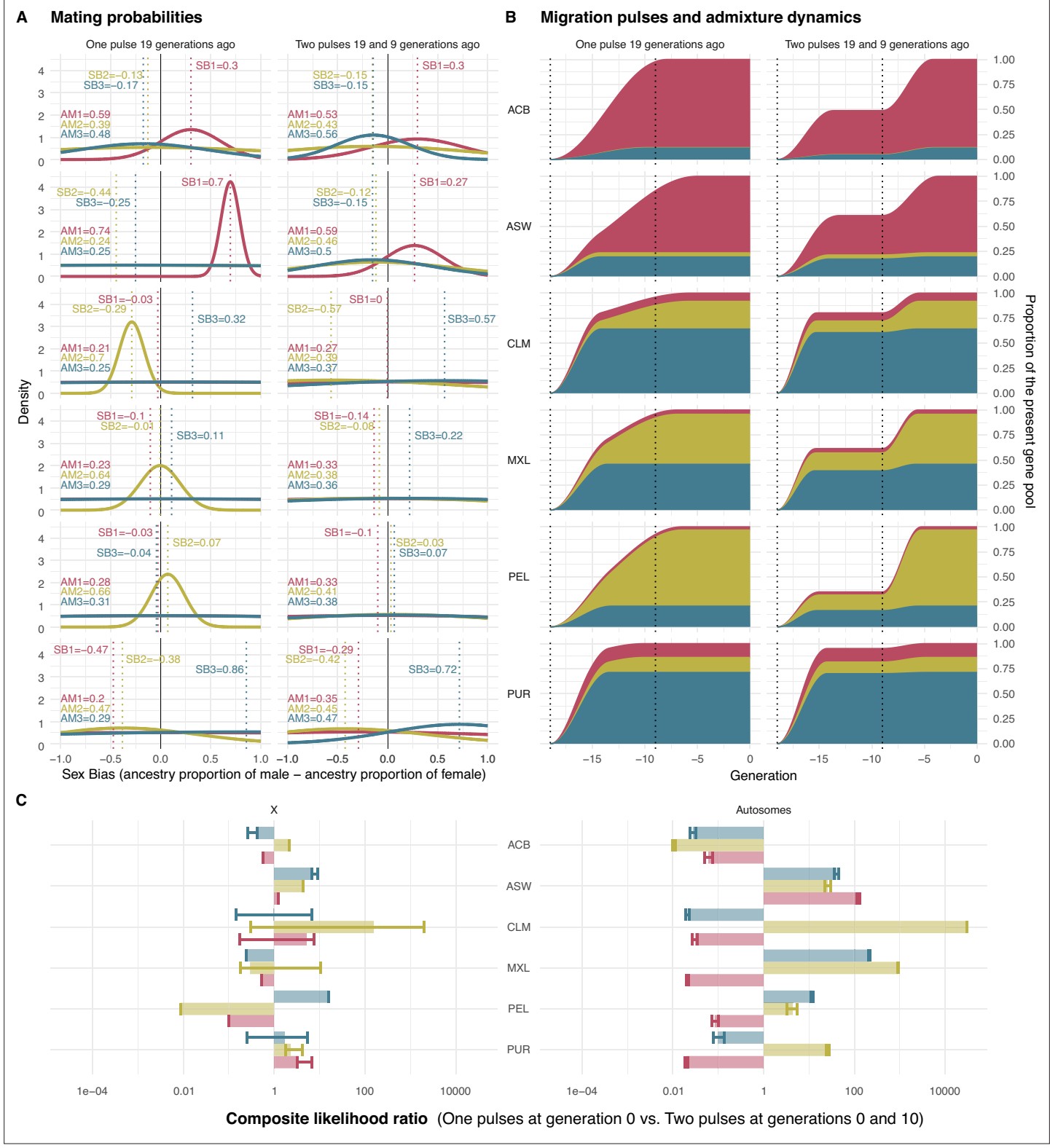

**Figure 4.** Mating probabilities, migration pulses and admixture dynamics. (**A**) Mating probabilities as a function of male and female proportions of each ancestry, for each population. (**B**) Migration pulses of each ancestry according to scenarios allowing One Pulse and Two Pulses for each population. The y-axis represents the cumulative increase in the ancestry-specific gene pool relative to the final ancestry proportions, at each generation. The ancestry proportions at generation 19 represent the observed ancestry proportions of each population in real data. The increase in the cumulative ancestry-specific gene pool is defined by gene flow rate (GFR), while the slope of the increase is represented inversely proportional to assortative mating (AM).

*Figure 4 continued on next page*

*Figure 4 continued*

(**C**) Composite likelihood ratio comparing Two Pulses model vs. One Pulse model, for each ancestry for both the X chromosome and the autosomes. In this plot, positive values show a higher likelihood of the Two Pulses model based on the fit of the real fragment lengths in the distribution of fragment lengths of simulated data under the AM and sex bias (SB) parameters predicted by the neural network. Error bars define the 95% CI obtained by bootstrapping the tract lengths profile.

The online version of this article includes the following figure supplement(s) for figure 4:

**Figure supplement 1.** Correlation between simulated assortative mating (AM) values and predicted gene flow rate (GFR) values (Simulation only for PUR, due to large computational requirements).

**Figure supplement 2.** Correlation between simulated gene flow rate (GFR) values and predicted assortative mating (AM) values (Simulation only for PUR, due to large computational requirements).

**Figure supplement 3.** Prediction of assortative mating in the One Pulse model using the continuous ancestry tract lengths profile from Gnomix or RFMix.

**Figure supplement 4.** Prediction of sex bias in the One Pulse model using the continuous ancestry tract lengths profile from Gnomix or RFMix.

**Figure supplement 5.** Prediction of assortative mating in the Two Pulses model using the continuous ancestry tract lengths profile from Gnomix or RFMix.

**Figure supplement 6.** Prediction of sex bias in the Two pulses model using the continuous ancestry tract lengths profile from Gnomix or RFMix.

**Figure supplement 7.** Prediction of gene flow rate in the Two Pulses model using the continuous ancestry tract lengths profile from Gnomix or RFMix.

network trained to predict AM could predict AM from data with no assortative mating but only gene flow due to migration. We obtain strong correlations between simulated AM and predicted GFR, and between simulated GFR and predicted AM, which points that both demographic events could lead to similar admixture dynamics and would leave resembling genomic footprints in the population (*Figure 4—figure supplements 1 and 2*).

We next sought to test whether observed genomic data is more compatible with a One Pulse or Two Pulses migration model. To this aim, we calculated the composite likelihood ratio to compare the fit of simulated continuous ancestry tract length profile under the predicted values of AM and SB (and GFR for the Two Pulses model) to the empirical data. In ASW, CLM, MXL, and PEL populations the Two Pulses model has greater support than the One Pulse model. Results also show that, in all these populations, the Two Pulses model is more supported by the continuous ancestry tract length profile from the autosomal chromosomes than the profile from the X chromosome. (*Figure 4C*).

## Discussion

We tackle the analysis of the ancestry-related non-random mating driven by social structure through a mating model, which allows us to globally evaluate the forces that modulate population structure but also study the effect of these population dynamics at the individual level. Our results show evidence of ancestry-related sex bias and assortative mating in American admixed populations. In Latin Americans, the proportion of Native American ancestry of men and women shape the mating probabilities and, therefore, the genetic structure of the population. By contrast, in African Americans, the sub-Saharan African ancestry modulates mating. Below, we evaluate the performance of our pipeline in discerning between migration and assortative mating and we explore how the next steps could incorporate more complex admixture scenarios. Finally, we discuss the significance of these results and the importance of our approach in studying social stratification.

### Discerning geographical barriers and social barriers

The disentanglement of social barriers from geographical barriers is a major challenge to face when addressing the complexity of admixture. Non-random mating patterns associated with social structure and gene-flow after multiple migration pulses might leave similar footprints on the genome of a population. In our analyses, the admixture dynamics are modeled either by ancestry-related assortative mating (both in the One Pulse model and the Two Pulses model) or migration pulses (in the Two Pulses model). In both cases, the ancestry modulating the admixture dynamics takes longer to homogenize across the individuals of the admixing population. We have shown that an earlier admixture process under higher assortative mating might be interpreted as a later migration pulse under random mating or lower assortative mating. This has important implications for admixture dating

methods that assume random mating, which can underestimate the generations from the admixture event if the admixture process started earlier and took place under assortative mating.

Despite the similarity of the genomic footprint left by both demographic events the distinction between both scenarios has important implications for understanding social processes. The continuous ancestry tract length profile issued from our mating model is different from that issued from an admixture model where the admixed population only receives constant gene-flow from migration of non-admixed source populations to model the population structure. In the latter approach, as defined in *Goldberg and Rosenberg, 2015*; *Laurent et al., 2022*, after migration the non-admixed source populations introduce full-length non-recombined chromosomes with a single ancestry in the admixed populations and shape an identifiable pattern. In our approach, the continuous ancestry tract length profile is capturing this pattern to discern between ancestry-related assortative mating and gene-flow due to migration. Although this pattern characterized by low or non-recombined continuous ancestry tracts in a few individuals could remain hidden in a population mean continuous ancestry tract length profile, it should be detectable with an individual-based one.

Alternatively, a population structure correlated with genetic ancestry could also be modeled with an island model starting from non-admixed panmictic subpopulations with migration rates between them split by sex. Our approach resembles this model but considers a more realistic population continuously structured by genetic ancestry, rather than discrete (although permeable) subpopulation panmictic demes. In this sense, we address population structure through a mating model, which helps to focus the discussion on the effect at individual level of the dynamics that stratify the population.

## A versatile mating model to accommodate a wider range of admixture scenarios

The different patterns inferred in autosomal and X chromosomes suggest that some of the complexity of admixture is still not explained by our models. Inferences based on autosomal chromosomes have a greater support for a demographic scenario involving multiple migration events than inferences on the X chromosome. Taken together, these trends suggest a need to expand the model by adding either the possibility of sex-biased migrations or changes in SB and AM through time. In addition, African-American populations might have a complex genetic history involving on one hand male-biased sub-Saharan migration and on the other hand an admixture female-biased in the sub-Saharan ancestry. However, our current model can only accommodate this demographic scenario with a single SB parameter, and the results regarding these populations should be interpreted with caution.

The flexibility of our methodology provides the basis to disentangle more complex scenarios of admixture in a further approach. The plasticity of forward-in-time simulators allows for additional complexity in the admixture model, by modifying the AM and SB parameters through time and modeling multiple migration pulses and population growth.

Machine learning has the potential to infer a large number of parameters issued from complex models of admixture. However, this step likely implies a shift from a population-based continuous ancestry tract length profile to an individual-based one, which needs to be linked to a redesign of the architecture of the neural network. Other architectures such as convolutional neural networks and generative models have been recently deployed to infer introgression, structure, admixture proportions, and post-admixture selection from population genomic data (*Gower et al., 2021*; *Wang et al., 2021*; *Meisner and Albrechtsen, 2022*; *Hamid et al., 2022*). We caution that local ancestry analysis is a sensitive up-stream step to our method. Particularly, a wrong ancestry prediction in isolated windows that breaks up longer continuous ancestry tracts can substantially affect the tract lengths profile. This might happen despite a high proportion of windows with the genetic ancestry accurately predicted that translates to an overall good performance of the local ancestry method.

## Social stratification by racial and gender hierarchies

The ultimate aim of our approach is to infer social stratification in the Americas from the analysis of the population genetic structure. To this end, we model ancestry-related assortative mating mediating the extent of the effect of ancestry-related sex bias. We infer how assortative mating and sex bias shape the mating dynamics of the population and we evaluate how the dimensions of inequality associated to these parameters impact at individual level, thanks to the mating model framework.

We have defined this model from an intersectional perspective, which understands that racial, gender and class hierarchies are mutually constituted (*Davis, 1981*; *Gonzalez, 1984*; *Hooks, 1984*; *Crenshaw, 1989*; *Crenshaw, 1991*; *Collins, 1990*; *Carneiro, 1995*; *McCall, 2005*; *Hancock, 2007*; *Viveros Vigoya, 2016*). Besides, from Decolonial Feminism, authors have stressed the significance of the frame of the European Colonization in redefining the concept of gender in America and have pointed to sexuality and mating as one of the contexts where racial, gender, and class inequalities manifest with more visibility (*Stolcke, 1992*; *Lugones, 2007*; *Lugones, 2008*; *Viveros Vigoya, 2009*; *Viveros Vigoya, 2016*). Specifically, in our approach, we consider that racial stratification intensifies gender inequalities during mating. At the individual level, we conceive that the effects of these social hierarchies on mating depend on the relative position of the subject in racial and gender axes of inequality respect to the position of the other individuals of the population, and therefore on the context (*Yuval-Davis, 2006*; *Anthias, 2013*; *Jorba and Rodó-Zárate, 2019*).

In this framework, we tackle the hypotheses about the role of *mestizaje/mestiçagem* in the inequality of a society developed by *Wade, 2017*; *Wade et al., 2020* through a genetic study. Wade states that *mestizaje/mestiçagem* is a highly ambivalent discourse and set of practices, as it promotes and facilitates interactions across hierarchical differences of 'race,' class, and gender, but simultaneously reinforces those hierarchies. Studying the interaction of racial and gender hierarchies together, our approach allows us to investigate the dynamics of *mestizaje/mestiçagem* and challenge the oversimplified understanding that greater mixture translates into a decrease of discrimination and inequality. As such, high levels of mixture cannot be understood as a sign of equality or low discrimination if it takes place with strong gender-biased patterns. Instead, this scenario provides evidence for a deep interaction between racial and gender hierarchies that shapes the social structure of the population.

In the specific case of the Latin American populations considered here, the prejudices and biases related to gender associated with Native American ancestry constrain the dynamics within the society and determine the mating, shaping the population genetic structure. In CLM and PUR populations, females with a higher proportion of Native American ancestry tend to mate with males of lower Native American Ancestry. Interestingly, the populations that in the Two Pulses model present a higher migration pulse from Native American populations 9 generations ago (MXL and PEL) do not present a significant negative sex bias. This additional gene-flow probably took place under weaker sex-biased patterns and shifted the observed *SB* parameter towards non-negative values. This is consistent with historical records reporting an increased pressure on Native populations to culturally assimilate towards a white/mestizo norm starting in the 19[th] century, which spurred internal migrations to urban areas and the loss of indigenous languages in places like Mexico and Peru (*Viqueira, 2010*; *Telles, 2014*).

Our results support the idea that *Mestizaje/mestiçagem* operates through racial and gender hierarchies and it is accompanied by a gradual dilution of the sociocultural elements associated with non-European genetic ancestries into the admixing population.

## Conclusion

To have a broader perspective of how racial and gender hierarchies operate across American societies, further approaches should analyze a wider dataset with more diverse, representative and carefully sampled populations. Specifically, the inclusion of socioeconomic variables in the sampling would allow us to evaluate how class hierarchies interrelate with racial and gender hierarchies. In addition, the transition to a more complex admixture model that monitors changes in the AM and *SB* parameters and includes sex unbalanced migrations should provide the possibility to evaluate how racial and gender hierarchies have changed through history in different regions of the Americas.

In conclusion, an interdisciplinary approach that incorporates up-to-date insights from social sciences is essential to conceptualize population genetic models that aim to evaluate genetic structure driven by social stratification. Further studies might expand the analysis of population stratification by exploiting the full potential of machine learning in population genetics. An intersectional perspective that jointly addresses the effects of racial, gender, and class hierarchies on population structure will be key to understanding the genetics of the admixing populations of the Americas.

## Materials and methods

### Use of genetic ancestry categories

Human genetic diversity is a continuum and does not show discrete groups of individuals. Therefore, race is not a proxy for human genetic variation. Similarly, genetic ancestry labels are not distinct categories in which the genetic diversity of a population can be intrinsically grouped. Instead, the inference of genetic ancestry should be used in a hypotheses-driven approach framed in a specific time horizon (*Mathieson and Scally, 2020*; *Coop, 2022*; *Lewis et al., 2022*). In any case, researchers should disclose the process by which they selected and assigned group labels and the rationale for any grouping of samples (*National Academies of Sciences, Engineering, and Medicine, 2023*).

Here, we work with genetic ancestry categories that refer to the part of the genome inherited from each of the three populations involved in the admixture process in the Americas in the particular framework of the European colonization (Native American, European, and sub-Saharan African). We use these genetic ancestry categories because we hypothesize that they are correlated with the socially constructed racial and gender hierarchies that shape the population structure in the Americas.

### A mating model defined by ancestry-related assortative mating and sex bias

We derived a mechanistic model where AM and SB parameters constrain the mating probabilities as a function of the difference in the ancestry proportion between male and female, for each ancestry. We assume that the probability of mating follows a normal distribution where SB for each ancestry is defined as the expected value of the difference in the ancestry proportion within mating couples, while *AM* is modeled as being inversely proportional to its variance (*Figure 1*).

### Model definition

Consider an admixed finite population from $S$ isolated source populations comprised of $F$ females and $M$ males. Assume that for each individual $i$ we have a vector of inferred ancestry proportions $\vec{a}_i$ for each source population $s$, so that $\sum_{s=1}^{S} a_i^{\{s\}} = 1$. We consider a random variable for mating $L$ as a realization of the event $l_{f,m}$ between a female $f$ and male $m$.

We calculate the probability of mating between a female $f$ and male $m$ as

$$P(L = l_{\{f,m\}}|\vec{a_f}, \vec{a_m}) \quad = \frac{1}{2}(P(l_f|\vec{a_f}) \cdot P(l_{\{f,m\}}|l_f, \vec{a_f}, \vec{a_m})) + $$
$$+ \frac{1}{2}(P(l_m|\vec{a_m}) \cdot P(l_{\{f,m\}}|l_m, \vec{a_f}, \vec{a_m})) \tag{1}$$

where $P(l_f|\vec{a_f})$ and $P(l_f|\vec{a_m})$ are the probabilities of either female or a male to start a mating event that will have one child as the outcome.

$P(l_{\{f,m\}}|l_f, \vec{a_f}, \vec{a_m})$ is the probability of a female to mate with a male given the ancestry proportions of both female and male, once the female has already been chosen to initiate the mating.

$P(l_{\{f,m\}}|l_m, \vec{a_f}, \vec{a_m})$ is, therefore, the probability of a male to mate with a female given the ancestry proportions of both female and male, once the male has already been chosen to initiate the mating.

### Mating probability of a couple

In the most basic model, all individuals can be assumed to have the same probability to start the mating event independent of their ancestry:

$$P(l_f|\vec{a_f}) = \frac{1}{F} \tag{2}$$

$$P(l_f|\vec{a_m}) = \frac{1}{M} \tag{3}$$

Once either a female or a male initiates the mating, the mating probability of each possible couple is defined as a function of the ancestry proportions of this individual and the ancestry proportions of each individual of the other sex ($P(l_{\{f,m\}}|l_f, \vec{a_f}, \vec{a_m})$ or $P(l_{\{f,m\}}|l_m, \vec{a_f}, \vec{a_m})$ for respectively a male or a female that initiates the mating).

This probability is described by a multivariate normal distribution defined by a mean vector $\mu$, related to SB, and covariance matrix $\Sigma$, related to AM. This multivariate normal distribution consists of S variables defined as $a_m^{\{s\}} - a_f^{\{s\}}$, related to each ancestry $s$. They account for the difference of the ancestry proportion in each mating couple. The final mating probability for an individual and a possible mate is relative to the sum of all the probabilities for all the possible mates for this individual:

$$P(l_{\{f,m\}}|l_f, \vec{a_f}, \vec{a_m}) = \frac{\mathcal{N}(a_f^{\{s\}} - a_m^{\{s\}}|\mu, \Sigma)}{\sum_{j=1}^{M} \mathcal{N}(a_f^{\{s\}} - a_{m_j}^{\{s\}}|\mu, \Sigma)} \tag{4}$$

and:

$$P(l_{\{f,m\}}|l_m, \vec{a_f}, \vec{a_m}) = \frac{\mathcal{N}(a_f^{\{s\}} - a_m^{\{s\}}|\mu, \Sigma)}{\sum_{i=1}^{F} \mathcal{N}(a_{f_i}^{\{s\}} - a_m^{\{s\}}|\mu, \Sigma)} \tag{5}$$

where $\mu$ is the vector of the expected means of the ancestry proportion differences (i.e. $\mathbb{E}(a_m^{\{s\}} - a_f^{\{s\}})$ for ancestry $s$) which defines SB for each ancestry. The diagonal of $\Sigma$ is the vector $\sigma_S^2$, where the variance $\sigma^2$ of $(a_m^{\{s\}} - a_f^{\{s\}})$ is inversely proportional to AM for each ancestry (**Figure 1**).

The sum of the mean vector (i.e. the sum SB parameters for all the ancestries) is zero ($\sum_{s=1}^{S} \mu^{\{s\}} = 0$). In addition, $\Sigma$ is not full rank ($|\Sigma| = 0$). Consequently, the multivariate density function can be represented with $S - 1$ dimensions, which has $\frac{(S-1)^2 + (S-1)}{2}$ independent parameters.

## The case of three ancestries, S=3

When S=3, the multivariate normal distribution is equivalent to a two-dimensional multivariate normal distribution, which has five independent parameters (2 in $\mu$ and 3 in $\Sigma$):

$$\mu = \begin{bmatrix} \mu_1 \\ \mu_2 \end{bmatrix} \tag{6}$$

$$\Sigma = \begin{bmatrix} \sigma_1^2 & Cov_{1,2} \\ Cov_{1,2} & \sigma_2^2 \end{bmatrix} \tag{7}$$

where $Cov_{1,2}$ can be defined by the variances of the three-dimensional multivariate normal distribution, including $\sigma_3^2$:

$$Cov_{1,2} = \frac{\sigma_3^2 - (\sigma_1^2 + \sigma_2^2)}{2}, \tag{8}$$

The mating model for three ancestries is set by sex bias for ancestries 1 and 2 (SB$_1$, SB$_2$) and the assortative mating for ancestries 1, 2, and 3 (AM$_1$, AM$_2$, AM$_3$). Therefore, SB for ancestries $s \in \{1, 2\}$ is defined as follows:

$$SB_s = \mu_s \tag{9}$$

and AM for ancestries $s \in \{1, 2, 3\}$:

$$AM_s = 1 - \frac{\log_3(\sigma_s^2) + 4}{7} \tag{10}$$

This arbitrary parameterisation has been chosen in order for the the assortative mating parameter AM$_s$ to cover the full spectrum of meaningful values taking values from 0 to 1 on a logarithmic scale, where 0 is random mating and 1 is very strong assortative mating.

## Simulations

We performed 10,000 simulations per population (CLM, MXL, PEL, PUR, ACB, ASW from 1000 genomes) following the mating model described above using SLiM (**Haller et al., 2019**). In each

simulation, AM and SB parameters (and GFR for the Two Pulses model) were independently sampled from a uniform distribution. We simulated the 22 autosomal chromosomes and the X chromosome for each individual. We tracked their real local ancestry by recording the source population from which each genomic fragment is inherited. At each generation, we simulated mating based on the mating model as a function of the genetic ancestry proportions of the individuals. After the mating of two individuals, we simulated recombination in the gametogenesis of the offspring and the progressive break down of the continuous ancestry tracts, using the local recombination probabilities from the genetic map provided in *Delaneau et al., 2019*. We ran a total of 19 generations, mirroring the time range from the beginning of the colonization to present day. For computational purposes, we scaled down by a factor of 1000 the lengths and recombination rates of the genome.

For the One Pulse demographic model, we simulated a constant population size of 1000 and we set initial gene-flow proportions equivalent to the observed genetic ancestry proportions for each population *Table 1*. For the Two Pulses model, we split the same size of migrant population from each source population in two migration waves at generation 0 and 10 as a function of the GFR parameter.

## The continuous ancestry tract lengths profile

The continuous ancestry tract length profile is a statistic that is commonly used to date admixture events, assuming random mating. However, here we exploited the information summarised with this statistic to assess the gene-flow related to both migration and assortative mating. In addition, we included the population continuous ancestry tract length profiles of both autosomes and X chromosome to provide to the neural network information that can be used to predict sex bias. While both sexes contribute equally to the autosomal genepool, females and males contribute 2/3 and 1/3, respectively, to the X chromosome genepool. This asymmetric inheritance between autosomes and X chromosome combined with local ancestry information is highly informative of the complexity of sex-biased admixture histories (*Goldberg and Rosenberg, 2015*).

**Table 1.** Average proportions percentage (and 95% CI) of genetic ancestry for each population, inferred after a local ancestry analysis with RFMix.

| Population | Ancestry | Aut | X |
|---|---|---|---|
| ACB | AFR | 88.3 (75.9, 97.4) | 94 (72.6, 99.6) |
| ACB | NAT | 0.1 (0, 0.2) | 0.1 (0, 1) |
| ACB | EUR | 11.8 (2.8, 24.2) | 5.4 (0, 25.7) |
| ASW | AFR | 76.9 (49.9, 91.4) | 77.5 (39.7, 99.6) |
| ASW | NAT | 1.5 (0, 13.2) | 3.7 (0, 27.6) |
| ASW | EUR | 21.7 (8.7, 41.2) | 18.3 (0, 54.2) |
| CLM | AFR | 8.2 (1.4, 23.5) | 8 (0, 43.4) |
| CLM | NAT | 26.8 (10.5, 43.4) | 40.1 (4.8, 81.4) |
| CLM | EUR | 65.1 (41.6, 86.8) | 51.5 (12.5, 91.9) |
| MXL | AFR | 4.4 (1, 7.7) | 5.2 (0, 29.7) |
| MXL | NAT | 49.3 (23.3, 87.7) | 61.2 (11.4, 99.5) |
| MXL | EUR | 46.5 (11.5, 72.9) | 33.2 (0, 86.4) |
| PEL | AFR | 3.2 (0.1, 14) | 5.1 (0, 32.9) |
| PEL | NAT | 75.4 (50.5, 95.4) | 81.9 (37.3, 99.6) |
| PEL | EUR | 21.6 (4, 43) | 12.5 (0, 47.8) |
| PUR | AFR | 13.8 (4.8, 24.2) | 15.4 (0.3, 56.6) |
| PUR | NAT | 14.4 (7, 21.2) | 21.2 (0.4, 53.3) |
| PUR | EUR | 72 (56.1, 86.3) | 63 (21.9, 88.2) |

**Table 2.** Average proportions (%) of genetic ancestry for each population, inferred after a local ancestry analysis with Gnomix.

| PopulatioAn | ancestry | Aut | X |
| --- | --- | --- | --- |
| ACB | AFR | 88.5 (75.8, 97.6) | 93.6 (70.9, 99.5) |
| ACB | NAT | 0.1 (0, 0.5) | 0.1 (0, 0.4) |
| ACB | EUR | 11.4 (2.3, 23.7) | 5.9 (0, 28.6) |
| ASW | AFR | 77 (47.7, 91.7) | 76.2 (13.6, 99.5) |
| ASW | NAT | 3.8 (0, 24.2) | 5.7 (0, 55.5) |
| ASW | EUR | 19.3 (6.8, 36) | 17.6 (0, 65.3) |
| CLM | AFR | 7.9 (0.9, 24.4) | 7.8 (0, 51.6) |
| CLM | NAT | 27.1 (10, 44.9) | 39.1 (3, 84.8) |
| CLM | EUR | 65(41, 89) | 52.6 (8.4, 95.9) |
| MXL | AFR | 4.2 (0.8, 8.2) | 5.1 (0, 28.6) |
| MXL | NAT | 50(23, 87) | 60.2 (9.1, 99.5) |
| MXL | EUR | 45.8 (11.3, 73.9) | 34.1 (0, 87) |
| PEL | AFR | 3.1 (0, 13.5) | 5.1 (0, 33.2) |
| PEL | NAT | 77 (52.2, 98.2) | 81.6 (32.5, 99.5) |
| PEL | EUR | 19.9 (1.6, 40.6) | 12.8 (0, 51.3) |
| PUR | AFR | 13.5 (4.8, 25.1) | 15 (0, 56.1) |
| PUR | NAT | 14.3 (6.7, 21.5) | 20.4 (0, 60.7) |
| PUR | EUR | 72.2 (57.3, 87.9) | 64.1 (23.6, 93.1) |

We calculated the continuous ancestry tract length profile on simulated data, for each individual, by counting the number of tracts for each length bin, greater than or equal to the lower threshold and lower than the upper threshold, defined by a vector of break points $b$ in a logarithmic scale, in centiMorgan (cM): $\{b_k\}_{k=1}^{21} = \frac{2^{\frac{k+1}{2}}}{10}$. These breakpoints define a total of 22 length windows, which is a compromise of the RFMix resolution in the local ancestry analysis (0.1 cM) and a limited number of windows. Then, we obtain the continuous ancestry tract length profile for each individual. Finally, we perform the mean across the individuals of the same population as the permutation-invariant function to use it as input of he neural network. For real empirical data, we run a local ancestry analysis to split the genome of each individual into the fragments inherited from Native American, European, and sub-Saharan African ancestries to obtain the length of the continuous ancestry tracts (*Figure 2A*). To do this, we performed an RFMix analysis with RFMix v1.5.4 (*Maples et al., 2013*) with the following options: -w 0.1 G 19 -e 3. We used as target populations the six admixed populations of the Americas present in the 1000 genomes data (African Caribbeans in Barbados, ACB; African Ancestry in SW USA, ASW; Colombians in Medellín, CLM; Mexicans in Los Angeles, MXL; Peruvians in Lima, PEL; and Puerto Ricans in Puerto Rico, PUR) using the 30 x coverage data 1000 (*Auton et al., 2015*; *Byrska-Bishop et al., 2022*).

To create three reference populations we first combined 1000 genomes with HGDP genomes (*Bergström et al., 2020*). We ran an unsupervised $k = 3$ ADMIXTURE analysis (*Alexander et al., 2009*), from which we used the individuals with a proportion higher than 0.99 of one of the ancestries as reference populations for the RFMix analyses. For Native American ancestry (NAT): 6 Colombian, 12 Karitiana, 13 Maya, 13 Pima, 8 Surui, 2 MXL, and 19 PEL (these PEL and MXL individuals are also included in the target population). For European ancestry (EUR): 23 Basque, 12 BergamoItalian, 28 French, 15 Orcadian, 28 Sardinian, 8 Tuscan, 98 CEU, 91 GBR, 98 IBS. For Sub-Saharan African ancestry (AFR): 8 BantuKenya, 8 BantuSouthAfrica, 22 Biaka, 21 Mandenka, 13 Mbuti, 6 San, 22 Yoruba, 3 ACB, 1 ASW,

**Table 3.** Estimated Mean (and 95 %CI) of the mating parameters for the One Pulse Model, using the continuous ancestry tract lengths profile obtained from RFMix as input to 1000 trained neural networks.

| Population | CLM | MXL | PEL | PUR | ASW | ACB |
|---|---|---|---|---|---|---|
| AM1 | 0.21 (0.17, 0.26) | 0.23 (0.19, 0.28) | 0.28 (0.25, 0.31) | 0.2 (0.13, 0.28) | 0.74 (0.64, 0.81) | 0.59 (0.48, 0.69) |
| AM2 | 0.7 (0.64, 0.76) | 0.64 (0.55, 0.71) | 0.66 (0.58, 0.74) | 0.47 (0.33, 0.62) | 0.24 (0.18, 0.31) | 0.39 (0.35, 0.44) |
| AM3 | 0.25 (0.2, 0.31) | 0.29 (0.24, 0.33) | 0.31 (0.27, 0.34) | 0.29 (0.21, 0.39) | 0.25 (0.19, 0.32) | 0.48 (0.41, 0.59) |
| SB1 | −0.03 (-0.28, 0.16) | −0.1 (-0.31, 0.1) | −0.03 (-0.18, 0.15) | −0.47 (−0.64, −0.25) | 0.7 (0.56, 0.81) | 0.3 (0.2, 0.4) |
| SB2 | −0.29 (−0.43, −0.17) | −0.01 (-0.15, 0.13) | 0.07 (-0.23, 0.26) | −0.38 (−0.54, −0.23) | −0.44 (−0.6, −0.31) | −0.13 (−0.23, −0.02) |
| SB3 | 0.32 (0.18, 0.49) | 0.11 (-0.03, 0.25) | −0.04 (-0.17, 0.13) | 0.86 (0.68, 1.01) | −0.25 (−0.41, −0.1) | −0.17 (−0.27, −0.08) |

**Table 4.** Estimated Mean (and 95 %CI) of the mating parameters for the One Pulse Model, using the continuous ancestry tract lengths profile obtained from Gnomix as input to 1000 trained neural networks.

| population | CLM | MXL | PEL | PUR | ASW | ACB |
|---|---|---|---|---|---|---|
| AM1 | 0.22 (0.18,0.26) | 0.26 (0.22,0.31) | 0.27 (0.24,0.31) | 0.16 (0.12,0.21) | 0.51 (0.33,0.65) | 0.51 (0.39,0.64) |
| AM2 | 0.45 (0.32,0.6) | 0.52 (0.38,0.64) | 0.48 (0.33,0.62) | 0.35 (0.24,0.49) | 0.26 (0.23,0.3) | 0.36 (0.32,0.4) |
| AM3 | 0.31 (0.24,0.44) | 0.34 (0.28,0.42) | 0.31 (0.27,0.37) | 0.23 (0.16,0.32) | 0.32 (0.26,0.39) | 0.4 (0.35,0.46) |
| SB1 | −0.01 (-0.24,0.2) | 0.17 (-0.04,0.41) | 0.12 (-0.1,0.36) | −0.08 (-0.28,0.08) | 0.01 (-0.35,0.31) | −0.08 (-0.29,0.11) |
| SB2 | −0.21 (−0.39,,−0.06) | −0.21 (−0.39,,−0.03) | −0.36 (−0.59,,−0.06) | −0.36 (−0.5,,−0.2) | −0.04 (-0.22,0.25) | 0.02 (-0.1,0.19) |
| SB3 | 0.21 (0.07,0.4) | 0.04 (-0.1,0.18) | 0.23 (0.04,0.39) | 0.44 (0.25,0.62) | 0.03 (-0.15,0.23) | 0.05 (-0.09,0.21) |

**Table 5.** Estimated Mean (and 95 %CI) of the mating parameters for the Two Pulses model, using the continuous ancestry tract lengths profile obtained from RFMix as input to 1000 trained neural networks.

| population | CLM | MXL | PEL | PUR | ASW | ACB |
|---|---|---|---|---|---|---|
| AM1 | 0.27 (0.22,0.34) | 0.33 (0.23,0.42) | 0.33 (0.25,0.44) | 0.35 (0.26,0.44) | 0.59 (0.45,0.73) | 0.53 (0.42,0.63) |
| AM2 | 0.39 (0.28,0.5) | 0.38 (0.28,0.48) | 0.41 (0.32,0.52) | 0.45 (0.32,0.6) | 0.46 (0.35,0.58) | 0.43 (0.36,0.52) |
| AM3 | 0.37 (0.29,0.45) | 0.36 (0.25,0.47) | 0.38 (0.3,0.47) | 0.47 (0.38,0.56) | 0.5 (0.39,0.61) | 0.56 (0.43,0.7) |
| SB1 | 0 (-0.18,0.18) | −0.14 (-0.35,0.13) | −0.1 (-0.27,0.07) | −0.29 (−0.54,,−0.06) | 0.27 (-0.03,0.52) | 0.3 (0.18,0.43) |
| SB2 | −0.57 (−0.68,,−0.42) | −0.08 (-0.33,0.15) | 0.03 (-0.17,0.23) | −0.42 (−0.59,,−0.26) | −0.12 (-0.36,0.15) | −0.15 (-0.33,0.06) |
| SB3 | 0.57 (0.42,0.72) | 0.22 (-0.07,0.45) | 0.07 (-0.17,0.26) | 0.72 (0.5,0.93) | −0.15 (-0.42,0.17) | −0.15 (-0.32,0.05) |
| GFR1 | 0.01 (0.01,0.02) | 0 (0,0.01) | 0.03 (0.01,0.06) | 0.02 (0.01,0.04) | 0.49 (0.27,0.69) | 0.5 (0.32,0.71) |
| GFR2 | 0.58 (0.44,0.7) | 0.64 (0.47,0.76) | 0.79 (0.7,0.87) | 0.23 (0.11,0.35) | 0 (0,0.01) | 0.27 (0.14,0.43) |
| GFR3 | 0.06 (0.02,0.1) | 0.14 (0.07,0.26) | 0.22 (0.13,0.34) | 0.02 (0.01,0.03) | 0.11 (0.03,0.21) | 0.61 (0.45,0.74) |

99 ESN, 102 GWD, 45 LWK, 85 MSL, 107 YRI. To avoid biases introduced by haploid X chromosomes of males, we only used the X chromosome RFMix output from females to generate the continuous ancestry tract length profile. We considered a tract the concatenation of contiguous 0.1 cM fragments with maximum posterior probability of being inherited from one of the three ancestries. We used the same break points to count the fragments in each length bin used in the simulations: $\{b_k\}_{k=1}^{21} = \dfrac{2^{\frac{k+1}{2}}}{10}$ (in cM) to obtain the continuous ancestry tract length profile for each individual (*Figure 2B*). We bootstrapped the fragment length profile of each individual by resampling 1000 times with replacement each ancestry histogram and assuming its mean. Then, we computed the mean across individuals of the same population of the continuous ancestry tract length profile to have a single matrix for each population equivalent to the output of simulations used to train the neural network. We obtained a

**Table 6.** Estimated Mean (and 95 %CI) of the mating parameters for the Two Pulses model, using the continuous ancestry tract lengths profile obtained from Gnomix as input to 1000 trained neural networks.

| population | CLM | MXL | PEL | PUR | ASW | ACB |
|---|---|---|---|---|---|---|
| AM1 | 0.45 (0.38,0.54) | 0.54 (0.47,0.63) | 0.54 (0.44,0.64) | 0.49 (0.39,0.57) | 0.76 (0.69,0.83) | 0.61 (0.48,0.71) |
| AM2 | 0.51 (0.4,0.6) | 0.6 (0.51,0.68) | 0.68 (0.58,0.77) | 0.59 (0.48,0.71) | 0.62 (0.53,0.72) | 0.53 (0.42,0.65) |
| AM3 | 0.66 (0.57,0.74) | 0.64 (0.55,0.72) | 0.57 (0.46,0.69) | 0.66 (0.58,0.76) | 0.62 (0.53,0.71) | 0.57 (0.48,0.65) |
| SB1 | −0.06 (-0.29,0.15) | −0.27 (−0.44,,−0.12) | −0.14 (-0.37,0.09) | −0.04 (-0.28,0.2) | 0.05 (-0.15,0.25) | −0.18 (-0.41,0.08) |
| SB2 | −0.21 (-0.41,0.06) | 0.04 (-0.16,0.25) | 0.01 (-0.22,0.22) | −0.33 (−0.48,,−0.13) | −0.1 (-0.32,0.13) | 0.09 (-0.09,0.33) |
| SB3 | 0.27 (0.12,0.41) | 0.23 (0.02,0.44) | 0.13 (-0.18,0.46) | 0.37 (0.15,0.56) | 0.05 (-0.25,0.32) | 0.09 (-0.14,0.32) |
| GFR1 | 0.09 (0.04,0.15) | 0.05 (0.02,0.1) | 0.1 (0.04,0.2) | 0.07 (0.03,0.12) | 0.33 (0.18,0.46) | 0.63 (0.4,0.82) |
| GFR2 | 0.21 (0.14,0.32) | 0.33 (0.22,0.44) | 0.38 (0.22,0.54) | 0.18 (0.09,0.29) | 0.17 (0.08,0.27) | 0.34 (0.22,0.49) |
| GFR3 | 0.31 (0.18,0.46) | 0.21 (0.11,0.32) | 0.1 (0.04,0.17) | 0.25 (0.14,0.37) | 0.1 (0.05,0.17) | 0.11 (0.04,0.2) |

**Table 7.** Neural Network mean MSE and RFMix-Gnomix mean generalized variance (GV) after each modification of the continuous ancestry tract length profile in the One Pulse model.

The tract length profile has been modified either by dividing or not each value of the histogram by the total number of tracts in the Autosomes or in the X Chromosome, or by either removing or not the window corresponding to the shortest tracts. We have evaluated GV and the MSE reported in *Figure 3* in each case. GV is the determinant of the covariance matrix: $GV = var(RFMix) + var(Gnomix) - cov(RFMix, Gnomix)^2$ increases its value when the correlation between RFMix and Gnomix estimates is high, and the variances within Gnomix and RFMix estimates are low *Figure 4—figure supplements 5 and 6* show the scatter plots for all the mating parameters estimations.

| Windows | Scaling | Mean GV | Mean MSE |
|---|---|---|---|
| All windows | Divided by total sum of tracts | 8.34e-06 | 0.0297 |
| All windows | Raw | 1.07e-04 | 0.0295 |
| Without shortest tract window (<0.2 cM) | Divided by total sum of tracts | 4.44e-06 | 0.0303 |
| Without shortest tract window (<0.2 cM) | Raw | 1.03e-05 | 0.0295 |

three-dimensional matrix (22 length windows, three ancestries, two autosomal/X chromosome) that we use as the input to the trained neural network to predict the AM and SB parameters (and GFR for the Two Pulses model) (*Figure 2C*).

We also performed a Local Ancestry analysis with Gnomix, training a new model using the same reference populations as in the RFMix analysis (*Hilmarsson et al., 2021*). Like in the RFMix continuous ancestry tract length profile, we only used the X chromosome gnomix output from females. Once generated the tract lengths profile we estimated the mating parameters (*Table 2*, Tables 4 and 6, *Figure 2—figure supplements 1–6*). To evaluate possible biases in the mating paramaters estimation caused by errors in the Local Ancestry inference, we tested the performance of a transformed tract lengths profile for both RFMix and Gnomix. We modified the tract lengths profile by removing or not the window corresponding to the shortest tracts, or by dividing or not each value of the histogram by the total number of tracts in the Autosomes or in the X Chromosome. For each of the four combinations of the modifications of the tract lengths profile, we trained 1000 times the Neural Network to predict the mating parameters from either RFMix or Gnomix tract lengths profile (*Tables 3–8*, *Figure 2*, *Figure 4—figure supplements 3–7*). We then evaluated the performance of both estimations and the correlation between the results obtained from both tract lengths profiles. To do it, we analyzed the Generalized Variance of the bi-variate distribution, where the coordinates of each of the 1000 points are the predictions of a mating parameter (e.g. AM1) for one of the 1000 trained neural networks from both RFMix and Gnomix tract lengths profiles. The Generalized Variance is the determinant of the covariance matrix. Thus, it has lower values when the covariance between RFMix

**Table 8.** Neural Network mean MSE and RFMix-Gnomix mean generalized variance (GV) after each modification of the continuous ancestry tract length profile in the Two Pulses model.
The tract length profile has been modified either by dividing or not each value of the histogram by the total number of tracts in the Autosomes or in the X Chromosome, or by either removing or not the window corresponding to the shortest tracts. We have evaluated GV and the MSE reported in *Figure 3* in each case. GV is the determinant of the covariance matrix: $GV = var(RFMix) + var(Gnomix) - cov(RFMix, Gnomix)^2$ increases its value when the correlation between RFMix and Gnomix estimates is high, and the variances within Gnomix and RFMix estimates are low. *Figure 4—figure supplements 5–7* show the scatter plots for all the mating parameters estimations.

| Windows | Scaling | Mean GV | Mean MSE |
|---|---|---|---|
| All windows | Divided by total sum of tracts | 1.96e-05 | 0.0358 |
| All windows | Raw | 2.46e-04 | 0.0354 |
| Without shortest tract window (<0.2 cM) | Divided by total sum of tracts | 1.10e-05 | 0.0359 |
| Without shortest tract window (<0.2 cM) | Raw | 5.17e-05 | 0.0353 |

and Gnomix is higher and the variance within each of RFMix and Gnomix-based predictions is lower. Although all the mating parameters where estimated for the four modifications of the tract length profile, the profile with *All windows* and *Divided by total sum of tracts* was prioritized for downstream analysis, as it shows both low GV and low MSE values.

$$GV = var(RFMix) + var(Gnomix) - cov(RFMix, Gnomix)^2 \tag{11}$$

## Neural network

### Neural network architecture

We built a deep neural network comprised of four common fully-connected layers with 512, 256, 128, and 64 units, respectively, and ReLU activation functions. To avoid overfitting, we included a dropout layer with a rate of 0.2 after the last common layer. The network separates into five branches, each one for an independent parameter. Each branch forms a fully-connected layer with 32 units and ReLU activation functions followed by dropout with a rate of 0.2, and a final fully-connected output layer with a sigmoid activation function. There were five parameter branches for the One Pulse model ($AM_1$, $AM_2$, $AM_3$, $SB_1$, and $SB_2$) and three extra parameter branches for the Two Pulses model ($GFR_1$, $GFR_2$, $GFR_3$) (*Figure 2C*). In total the One Pulse model has 251,141 trainable weights and the Two Pulses model 263,819.

We used Adam as the optimizer and Mean Squared Error as the loss function. We rescale AM and SB from 0 to 1 to equally weight both parameters during learning. We trained the neural network for 40 epochs, a batch size of 64 with a validation split of 0.2 from the training and validation dataset. The training and validation dataset was a random 0.8 sample of the dataset comprising 10,000 matrices of the continuous ancestry tract lengths profile and we kept the remaining 0.2 for testing. We used Keras in Python to design and train the neural network (*Chollet, 2015*).

All the code is available at GitHub (copy archived at *Mas-Sandoval, 2024*).

## Acknowledgements

We thank Sarah Abel and Andres Ruiz-Linares for carefully reading the manuscript and for their insightful discussion regarding the implications of the findings. We would also like to thank Flora Jay for her valuable feedback on the methods.

MF and AMS are funded by The Leverhulme Research Project Grant RPG-2018–208. AMS is also funded by the European Research Council (ERC) under the European Union's Horizon 2020 research and innovation programme (grant agreement No 865356). SM is funded in part by the NIH grant R15HG011528. The content is solely the responsibility of the authors and does not necessarily represent the official views of the National Institutes of Health or other funding sources.

## Additional information

### Funding

| Funder | Grant reference number | Author |
| --- | --- | --- |
| Leverhulme Trust | RPG-2018-208 | Matteo Fumagalli<br>Alex Mas-Sandoval |
| Horizon 2020 | 865356 | Alex Mas-Sandoval |
| National Institutes of Health | R15HG011528 | Sara Mathieson |

The funders had no role in study design, data collection and interpretation, or the decision to submit the work for publication.

### Author contributions

Alex Mas-Sandoval, Conceptualization, Resources, Data curation, Software, Formal analysis, Validation, Investigation, Visualization, Methodology, Writing – original draft, Project administration, Writing – review and editing; Sara Mathieson, Supervision, Funding acquisition, Methodology, Writing – review and editing; Matteo Fumagalli, Conceptualization, Software, Supervision, Funding acquisition, Investigation, Methodology, Project administration, Writing – review and editing

### Author ORCIDs

Alex Mas-Sandoval (iD) http://orcid.org/0000-0002-1712-9404
Sara Mathieson (iD) https://orcid.org/0000-0002-0484-0838

### Decision letter and Author response

Decision letter https://doi.org/10.7554/eLife.84429.sa1
Author response https://doi.org/10.7554/eLife.84429.sa2

## Additional files

### Supplementary files

• MDAR checklist

### Data availability

The current manuscript uses already published data, so no data have been generated for this manuscript. The code used for the computational analyses is made available at the address stated in the methods.

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
