## [Editor Report]

In this important study, the authors develop a neural network to investigate assortative mating and sex-bias in admixed populations from the Americas. Applying their method to modern-day human genomes, they estimate sex-biased admixture and ancestry-based assortative mating. The evidence supporting their claims is solid, and their results will be of interest to population geneticists, anthropologists, and those interested in the history of the Americas.

---

## [Decision Letter]

**Decision letter after peer review:**

Thank you for submitting your article "The genomic footprint of social stratification in admixing American populations" for consideration by *eLife*. Your article has been reviewed by 2 peer reviewers, and the evaluation has been overseen by a Reviewing Editor and Molly Przeworski as the Senior Editor. The reviewers have opted to remain anonymous.

Full reviews are attached below, and should be carefully considered. Briefly, the most central areas to address include:

1) Assumption of known admixture timing: further simulations and/or empirical calculations to understand the role of this assumption, as well as careful discussion of the implications/choices, particularly because of the known impact of the timing of admixture on the variance of ancestry (which underlies the main inference).

2) Uncertainty: further estimates about the distributions/uncertainty/errors of key parameters of interest, specifically the level of sex bias and assortative mating. Consideration of alternative contributions to uncertainty are also important, particularly the role of potential errors in local ancestry inference on the X vs autosomes.

3) Contextualizing and discussing certain results (e.g. ASW mating patterns) that may be unintuitive and/or potentially conflict with previous publications. A more structured introduction and conclusion may help here.

*Reviewer #1 (Recommendations for the authors):*

– Regarding admixture dates, I suggest two analyses to tackle this possible issue. First, the authors may test how admixture date misspecification can bias AM estimation by using simulations as pseudo-empirical data, by setting an admixture time for the pseudo-empirical data that is largely different from the admixture time set for the simulations. Second, the authors could estimate admixture times in the observed data using the approach described in Zaitlen et al., Genetics 2017 (see also Korunes et al., G3 2022), and perform simulations with the corresponding admixture times, to train the neural network. A third, more demanding (but very interesting) option would be to co-estimate AM, SB and the admixture date using a deep neural network and the ancestry tract length distribution.

– Regarding local ancestry errors, could the authors compare the MSE of SB and AM estimation for pseudo-empirical simulated data where exact local ancestry is tracked (done already) and where local ancestry is inferred by RFMix from phased genotypes? This may be done for a small subset of models only, if the effect is minimal.

– Regarding the measure of uncertainty, could the authors report prediction intervals for their SB and AM estimates? This is particularly important, given the relatively low correlations obtained between estimated and true parameter values. Most intervals may include one.

– The authors report the composite likelihood ratio of the 1P vs. 2P models but do not test the significance of the ratio (probably because it is a composite likelihood) and do not assess how accurate is the choice between the two models. An option is to define a threshold for this composite likelihood ratio for which the probability to choose the true model is high, estimated from simulations and a confusion matrix.

– The Results section on the empirical data is somehow difficult to follow because no estimates of AM and SB are provided in the text. It is also missing some interpretation and discussion. If the authors are confident with their model choice, how do they interpret the higher fit for the 2P model for autosomes, relative to 1P model? Can the authors comment their results in light of previous findings (e.g., a positive SB value for ASW while negative in Bryc et al., AJHG 2015 and Ongaro et al., Genes 2021)?

*Reviewer #2 (Recommendations for the authors):*

I would love to see more data on the empirical ancestry inference, given that it's the input for the neural network and downstream analyses. It would be useful to have a figure that shows the continuous ancestry length profiles (and/or global ancestry proportion distributions) for all populations, potentially separated by males and females.

Could you include standard errors for the ancestry proportions in Table 1?

Can you add a supplemental table reporting the parameter estimates (either mean/variance or sex bias/assortative mating) from empirical data?

Can you specify what is being plotted in Figure 3? (I assume this is 95% confidence intervals and outliers but did not find this explicitly stated in the text.)

What was the rationale for a model where the second pulse occurs at generation 10?

Figure 4B is not referenced in the text and from the figure caption alone, it was unclear how the figure was constructed or how to interpret this result.

Are 1000 Genomes individuals used as reference populations in RFMix used again in downstream analyses? (In principle, I don't think there's a problem with doing so – was just unclear on the analysis pipeline.)

Does the extent of admixture impact the inference? For example, on average, 95.5% of the ancestry in ACB individuals comes from two ancestries, but ACB is still modeled as a 3-way admixture. It is promising that the mean squared error from simulations is similar between ACB and other populations, but I'm curious whether you've thought about modeling this as a 2-way admixture (and/or whether you'd expect the results to change if you did so).

I don't know what to make of the fact that strong assortative mating along one ancestry component (e.g. African ancestry in ASW) is not accompanied by strong assortative mating along any other ancestry component, especially in populations that are primarily by two out of the three ancestries. For example, in ASW, if males with high African ancestry are more likely to mate with females with low African ancestry, should this not automatically mean that males with low European ancestry are more likely to mate with females with high European ancestry?

Can you provide more details on the "joint parameter space" used to perform simulations and train the neural network?

---

## [Author Response]

Essential revisions:1) Assumption of known admixture timing: further simulations and/or emprical calculations to understand the role of this assumption, as well as careful discussion of the implications/choices, particularly because of the known impact of the timing of admixture on the variance of ancestry (which underlies the main inference).2) Uncertainty: further estimates about the distributions/uncertainty/errors of key parameters of interest, specifically the level of sex bias and assortative mating. Consideration of alternative contributions to uncertainty are also important, particularly the role of potential errors in local ancestry inference on the X vs autosomes.3) Contextualizing and discussing certain results (e.g. ASW mating patterns) that may be unintuitive and/or potentially conflict with previous publications. A more structured introduction and conclusion may help here.

We thank the editors and reviewers for appreciating the relevance of our work and for their constructive review and we submit a revised manuscript. We have addressed their main concerns in a comprehensive manner, focusing on the points raised by both reviewers and summarised by the editors. We aimed not only to respond to the reviewers' requests but also to incorporate the new analyses coherently into the main thread of the article:

We conducted new analyses that allow us to articulate a more robust and detailed discussion of how migration time and assortative mating inferences partially affect each other.

We used the Gene Flow Rate (GFR) and Assortative Mating (AM) parameters of our models to test this hypothesis. We show that later-than-considered migration events can be interpreted as assortative mating and that admixture dating assuming random mating can underestimate the time since admixture. In addition, this analysis contributes to discussing why both One Pulse and Two Pulses models predict similar admixture dynamics, either modelled by assortative mating or migration, and why they should be evaluated comprehensively.

We now provide 95% confidence intervals (CI) for each parameter obtained from the distribution of predicted parameters. In addition, we performed a local ancestry analysis with Gnomix, to evaluate how much the differences in the local ancestry inference can change the tract length profile and bias the estimation of the mating parameters. Moreover, we show that a simple normalisation step of the tract length profile reduces the sensitivity to the bias introduced by possible errors in the upstream local ancestry analysis without reducing the prediction accuracy.

We extended the discussion about the limitations to accommodate the demographic history of the studied populations only with the parameters of the current model and why the estimated values might differ from previous publications.

Besides these main points, we have individually addressed all the requests raised by the reviewers. Below, we detail all the analyses performed and we link the answers to reviewers to the manuscript location where we have incorporated the suggested changes.

Reviewer #1 (Recommendations for the authors):– Regarding admixture dates, I suggest two analyses to tackle this possible issue. First, the authors may test how admixture date misspecification can bias AM estimation by using simulations as pseudo-empirical data, by setting an admixture time for the pseudo-empirical data that is largely different from the admixture time set for the simulations. Second, the authors could estimate admixture times in the observed data using the approach described in Zaitlen et al., Genetics 2017 (see also Korunes et al., G3 2022), and perform simulations with the corresponding admixture times, to train the neural network. A third, more demanding (but very interesting) option would be to co-estimate AM, SB and the admixture date using a deep neural network and the ancestry tract length distribution.

We performed a new analysis to study how Assortative Mating (AM) inferences might be biased if the migration date is later than it has been considered, and how the time since migration can be underestimated if the admixture model does not account for assortative mating. As mentioned before, we trained a neural network by varying only the AM parameters (with no sex bias and a single migration pulse 19 generations ago). We then used this trained neural network to predict AM on simulated data obtained by varying only the migration size 9 generations ago (through GFR parameter), with no assortative mating and no sex bias. We also conducted a parallel experiment where we trained a neural network by varying only GFR (with no assortative mating and no sex bias). We used this trained neural network to predict GFR on simulated data obtained varying only AM, without migration nor sex bias.

We show the results of these analyses in figure 4-Supplement 1. These results suggest how the footprint left by either assortative mating or later-than-considered migration dates might be similar. For example, a Two Pulses scenario 19 and 9 generations ago with no assortative mating, where the gene flow rate in the second pulse is the 75% of total gene flow can also be interpreted as a single migration 19 generations ago and an AM value of around 0.55.

We also argue that both One Pulse and Two pulse models must be considered because they have a higher likelihood based on either X or Autosomes tract length profile, respectively. We also discuss how taking into account multiple migration pulses reduces AM values and how the resulting admixture dynamics resemble in both cases. This is detailed on line 264 in the results:

“To evaluate the similarity of the footprints left by either assortative mating or gene flow due to migration, we tested how a neural network trained to predict *GFR* could predict *GFR* from data with no gene-flow due to migration but only assortative mating. And, in parallel, we tested how a neural network trained to predict *AM* could predict *AM* from data with no assortative mating but only gene flow due to migration. We obtain strong correlations between simulated *AM* and predicted *GFR*, and between simulated *GFR* and predicted *AM*, which points that both demographic events could lead to similar admixture dynamics and would leave resembling genomic footprints in the population.”

and on line 327 in the discussion:

“In general, the admixture dynamics, either modelled by assortative mating or migration pulses, show a slower and progressive absorption of the Native American ancestry into the admixed population.”

– Regarding local ancestry errors, could the authors compare the MSE of SB and AM estimation for pseudo-empirical simulated data where exact local ancestry is tracked (done already) and where local ancestry is inferred by RFMix from phased genotypes? This may be done for a small subset of models only, if the effect is minimal.

We agree with the reviewers that a local ancestry of the simulated data would add soundness to our study. However, to constrain the computational cost of our analyses, our simulations do not generate genotype data but rather only the tracked true local ancestry. Additionally, as detailed in the text, we do not simulate explicit admixture of Native American, sub-Saharan and European populations by using vcf files as input. Nor do we explicitly simulate the full genetic history of the out-of-Africa and the peopling of America. Instead, we simulate 19 generations of admixture starting with three populations, whose local ancestry can be traced in the admixing population at each generation. We do it with specific flags in SLiM that track the origin population from which each genomic region of each admixed individual is inherited from. Our approach has the benefit of being computationally tractable in the context of training a deep learning algorithm. We acknowledge that future studies can investigate the potential of simulating full genomes for such inferences.

– Regarding the measure of uncertainty, could the authors report prediction intervals for their SB and AM estimates? This is particularly important, given the relatively low correlations obtained between estimated and true parameter values. Most intervals may include one.

As detailed above, we provide now the 95% CI for each parameter for both RFMix and Gnomix tract length profile.

– The authors report the composite likelihood ratio of the 1P vs. 2P models but do not test the significance of the ratio (probably because it is a composite likelihood) and do not assess how accurate is the choice between the two models. An option is to define a threshold for this composite likelihood ratio for which the probability to choose the true model is high, estimated from simulations and a confusion matrix.

We agree with the reviewers on the need to include a measure of uncertainty. We have included the confidence intervals of the composite likelihood, as explained above. All the confidence intervals at 95% are consistent with the point estimates, except for the sub-Saharan ancestry in the Colombian (CLM) population. As we argue in the results and the discussion, results from the autosomes and X chromosome support the Two Pulses and the One pulse model, respectively.

We argue that the admixture dynamics are similar in both models. In fact, the gene flow from Native American ancestry is homogenised to the gene pool of the admixed populations (by assortative mating or a second migration pulse) more progressively than for other ancestries. Then, we argue that the lack of concordance may indicate that a more complex model is needed, possibly by including sex-biased migrations besides sex-biased admixture and changing mating patterns through time.

– The Results section on the empirical data is somehow difficult to follow because no estimates of AM and SB are provided in the text. It is also missing some interpretation and discussion.

We provide the 95% CI for each parameter for both RFMix and Gnomix tract length profile in supplement Tables 2-5, as well as the bivariate distributions of predicted parameters in Figure 4 supplement 3-supplement 7. We have also extended the discussion of the results to facilitate the reading and interpretation.

If the authors are confident with their model choice, how do they interpret the higher fit for the 2P model for autosomes, relative to 1P model?

As discussed above, there is not a specific model choice, but both models are supported for either X or Autosomes. This is now comprehensively discussed, both in the results and, especially, in the discussion.

Can the authors comment their results in light of previous findings (e.g., a positive SB value for ASW while negative in Bryc et al., AJHG 2015 and Ongaro et al., Genes 2021)?

Rather than analyzing the sex bias only from the Autosomes/X chromosome ratio, our method infers the sex bias in mating from the tract length profile. ASW had a complex genetic history, probably involving both male sex-biased migrations and female sex-biased admixture, that also have likely changed through time. In our model, this is being modeled by a single parameter, and therefore the results should be interpreted with caution. We have incorporated a sentence in the subsection A versatile mating model to accommodate a wider range of admixture scenarios of the discussion, on line 380:

“In addition, African American populations might have a complex genetic history involving on one hand male-biased sub-Saharan migration and on the other hand an admixture female-biased in the sub-Saharan ancestry. However, our current model can only accommodate this demographic scenario with a single sex-bias parameter, and the results regarding these populations should be interpreted with caution.”

Reviewer #2 (Recommendations for the authors):I would love to see more data on the empirical ancestry inference, given that it's the input for the neural network and downstream analyses. It would be useful to have a figure that shows the continuous ancestry length profiles (and/or global ancestry proportion distributions) for all populations, potentially separated by males and females.

We have included new figures that show the continuous ancestry tract length profile generated after a local ancestry with RFMix and Gnomix (Figure 2 supplement 5-supplement 6) We also provide the distribution of individual proportions of ancestry inferred with both methods (Figure 2 supplement 1 – supplement 4 and Table 1 and Table supplement 1).

Could you include standard errors for the ancestry proportions in Table 1?

We include now the 95% CI in Table 1 and Table supplement 1 for RFMix and Gnomix inferences. Figure 2 supplement 1-supplement 4 show the distribution of individual proportions of ancestry.

Can you add a supplemental table reporting the parameter estimates (either mean/variance or sex bias/assortative mating) from empirical data?

We have included new tables with this information. Tables supplement 2 – supplement 5 include the parameter estimates with the 95% confidence intervals. Figures 4 supplement 3supplement 7 show the scatterplot of the estimations for each of the parameters, for both Gnomix and RFmix, as well as the correlation between the estimation from the tract length profile obtained from both methods.

Can you specify what is being plotted in Figure 3? (I assume this is 95% confidence intervals and outliers but did not find this explicitly stated in the text.)

We have modified the caption of figure 3 and included the following sentence: “The boxplot shows the distributions of values across the 1000 trained neural networks. The box limits are the 25th and 75th percentiles and the points show the outliers 1.5 times the interquartile range above the 75th percentile and below the 25th percentile.”

What was the rationale for a model where the second pulse occurs at generation 10?

*We* introduced a Two-Pulse model to illustrate how the genomic footprint of assortative mating can be similar to that of multiple migration pulses. A consequence of this fact is that admixture dating assuming random mating can underestimate the generations from the admixture event if the admixture process started earlier and took place under assortative mating. We made explicit this reflection in the discussion on line 351:

“This has important implications for admixture dating methods that assume random mating, which can underestimate the generations from the admixture event if the admixture process started earlier and took place under assortative mating.”

There are two main reasons behind the choice of setting the second migration pulse at generation 10. Firstly, it is convenient to set the second pulse in the middle of the 19 generations since the admixture process started. Secondly, this date coincides with an increase of the transatlantic slave-trade at the end of the 18th century, and close to an increase of European migration in the 19th century. In future studies, we will focus on assessing the impact of varying all these parameters to include the true complexity of the admixture process.

Figure 4B is not referenced in the text and from the figure caption alone, it was unclear how the figure was constructed or how to interpret this result.

We have included a reference to this figure after the sentence on line 257: “Both models reflect similar admixture dynamics, where Native American and sub-Saharan African genetic components are gradually incorporated into Latin American and African American gene pools, respectively.” We apologize, it was wrongly linked to figure 1.

Are 1000 Genomes individuals used as reference populations in RFMix used again in downstream analyses? (In principle, I don't think there's a problem with doing so – was just unclear on the analysis pipeline.)

The reviewer is correct as two MXL individuals and 19 PEL individuals are included both as reference for the Native American and in the target population. We have added a comment on line 523 to make it clearer: For Native American ancestry (NAT): 6 Colombian, 12 Karitiana, 13 Maya, 13 Pima, 8 Surui, 2 MXL and 19 PEL (these PEL and MXL individuals are also included in the target population).

Does the extent of admixture impact the inference? For example, on average, 95.5% of the ancestry in ACB individuals comes from two ancestries, but ACB is still modeled as a 3-way admixture. It is promising that the mean squared error from simulations is similar between ACB and other populations, but I'm curious whether you've thought about modeling this as a 2-way admixture (and/or whether you'd expect the results to change if you did so).

From our perspective, it is important to consider that the populations of the Americas went through homologous demographic events as a consequence of European colonisation. Although ACB and ASW could also be modelled from a 2-way model, we consider it is important to discuss the genetic history of these populations in the same framework as all the American populations to then discuss the differences between them. In fact, when we discuss the need to include diverse, representative and carefully sampled populations we mean that instead of ASW, we should have a representative sample of the southwest of the US, not restricted to the individuals self-identified as African-American. Then it would be more informative to compare a urban South-West US sample with a urban Colombian sample and discuss the important differences in the Assortative Mating parameters that would probably emerge.

I don't know what to make of the fact that strong assortative mati+ng along one ancestry component (e.g. African ancestry in ASW) is not accompanied by strong assortative mating along any other ancestry component, especially in populations that are primarily by two out of the three ancestries. For example, in ASW, if males with high African ancestry are more likely to mate with females with low African ancestry, should this not automatically mean that males with low European ancestry are more likely to mate with females with high European ancestry?

From equation 8, it is derived that: 1AM1=2.Cov2,3+1AM2+1AM3 In the case of ASW, the assortative mating of sub-Saharan African ancestry (AM_1_) is high. This can be compensated not only with high values of AM_3_, but also with high values of AM_2_ or low values of Cov_2,3_.

Besides, because the mating patterns define the relative probabilities of mating of all the possible couples, the mating patterns do not define the mating probabilities by themselves but also depend on the ancestry proportions of all the possible couples.

Can you provide more details on the "joint parameter space" used to perform simulations and train the neural network?

On line 477, we changed the sentence of the simulations subsection as it follows:

“In each simulation, AM and SB parameters (and GFR for the Two Pulses model) were independently sampled from a uniform distribution.”